# Foundry-Processed Compact and Broadband Adiabatic Optical Power Splitters with Strong Fabrication Tolerance

**Can Ozcan \***, **J. Stewart Aitchison** and **Mo Mojahedi**

Department of Electrical and Computer Engineering, University of Toronto, Toronto, ON M5S 3G8, Canada; stewart.aitchison@utoronto.ca (J.S.A.); mojahedi@ece.utoronto.ca (M.M.)
* Correspondence: can.ozcan@mail.utoronto.ca

**Abstract:** Optical power splitters play a crucial role as the fundamental building blocks for many integrated optical devices. They should have low losses, a broad bandwidth, and a high tolerance to fabrication errors. Adiabatic optical power splitters inherently possess these qualities while being compatible with foundry processes well suited for mass production. The long device lengths of adiabatic power splitters, however, are a limiting factor to achieve compact device sizes, which must be reduced. Here, we used a polynomial taper profile optimization algorithm to design 1 × 2 and 2 × 2 adiabatic power splitters with significantly shorter lengths than their adiabatic counterparts. The best-performing 1 × 2 and 2 × 2 power splitters had 20 μm and 16 μm coupling lengths, respectively. Our designs had minimum feature sizes ranging from 140 nm to 200 nm, and our measurements averaged across nine different chips showed excellent consistency in performance for devices with 180 nm and 200 nm minimum features. Both the 1 × 2 and 2 × 2 adiabatic optical power splitters had excess losses less than 0.7 dB over a 100 nm bandwidth, with a standard deviation lower than 0.3 dB. Furthermore, our measurements showed splitting ratios within 50 ± 3% over a 130 nm bandwidth. We also demonstrated the design of 1 × 2 power splitters with arbitrary splitting ratios, where splitting ratios ranging from 50:50 to 94:6 were achieved with standard deviations between 2% and 6%.

**Keywords:** power splitters; optical power splitting; adiabatic tapers; taper profile optimization





## 1. Introduction

Power splitters are among the most-fundamental building blocks of complicated on-chip optical circuits. They are used in optical switches [1], modulators [2], programmable optical circuits [3], and multiplexers [4], to name a few. Ideal power splitters should have low losses, a broad bandwidth, a compact size, a high tolerance to fabrication errors, and small deviations from the intended power splitting ratio. In addition, it is desired that the minimum feature sizes of power splitters be large (>150 nm), so that they are suitable for production at complementary metal–oxide semiconductor (CMOS) foundries using mass production. Satisfying these criteria simultaneously has proven to be challenging. There have been a wide variety of reported implementations of on-chip optical power splitters with differing strengths and weaknesses when the above-mentioned performance metrics are considered.

Among the various types of optical power splitters, directional couplers can achieve low loss and arbitrary power splitting ratios [5]. Yet, they are typically wavelength-sensitive, making it difficult to realize a polarization-insensitive and broadband response [6]. This problem has been partially solved by replacing the straight directional couplers with curved waveguides [1,7], which yields a broader bandwidth. The inherent strong polarization sensitivity, on the other hand, is difficult to overcome. Alternatively, power splitters based on multimode interferometers (MMIs) work using the self-imaging principle [8] and can be designed for an arbitrary number of inputs and outputs [9], as well as arbitrary splitting ratios [10]. Generally, MMI power splitters are less sensitive to wavelength than directional

couplers, although the designed devices can have a large footprint. For example, as in the case of a 2 × 2 3 dB coupler designed to operate with the TE0 and TE1 modes, a device length of 86.5 μm has been reported [11], whereas, in the case of 1 × 4 and 1 × 8 power splitters, device lengths of 36 μm and 47.8 μm have been achieved, respectively [12]. More-compact devices and polarization-insensitive operation have been achieved by engineering the width and thickness of the MMI power splitters, which yielded a footprint of 1.5 × 1.8 μm$^2$ [13]. However, this device was designed for a 300 nm-thick Si layer, which is not commonly used by industrial foundries. More-successful implementations of MMI power splitters have been achieved by modifying the profile of the MMI region, yielding a very compact and low-loss operation [14,15]. One downside of MMI power splitters is that the profile of the device between the MMI region and the output waveguides changes abruptly. This abrupt change constitutes a source for reflections and imbalance in the output, where a slight imbalance in the output waveguides can cause imperfect splitting or increased losses, as we demonstrated a similar effect in previous work [16].

More recently, subwavelength gratings (SWGs) have been used to improve the performance of optical power splitters. For example, the bandwidths of directional coupler-based power splitters were increased via utilizing SWGs [17,18]. A variety of designs for SWG-assisted directional couplers have been reported with low losses and a broad bandwidth [19–22]. SWG-assisted MMI-based power splitters have also been designed to improve the bandwidth, losses, and footprint of traditional MMI power splitters [23,24]. Moreover, SWG-assisted Y-junction power splitters have shown good performance [25–27]. However, it is important to note that the above-referenced works reported devices with minimum feature sizes smaller than 135 nm, making them unsuitable for large-scale fabrication at CMOS foundries.

In addition to the aforementioned approaches, various inverse design algorithms have been used to optimize the device power splitting efficiencies, where the device is divided into pixels [28–31] or subwavelength-sized stripes [32]. However, these optimization approaches yield devices that only work with the TE mode and consist of many small-sized features, which can result in fabrication intolerance [33]. When these devices are designed with the considerations of foundry limitations, inverse-designed power splitters with minimum feature sizes as large as 200 nm can be achieved [34]. However, these devices suffer from high losses and large output variations. Topological photonic devices have been an alternative to inverse-designed photonic components with small features [35]. With topological optimizations, foundry-compliant designs can be achieved by putting heavy constraints on the design [36,37]. In addition, machine learning models have been used to correct photonic device design layouts prior to fabrication [38]. Yet, these devices typically suffer from over 0.5 dB losses, and there is little discussion in the literature on making them polarization-insensitive.

Unlike other power splitter implementations, adiabatic optical power splitters are inherently low-loss, broad bandwidth , tolerant to fabrication errors, and polarization-insensitive. These advantages can be realized even with devices with large minimum feature sizes, yet the resultant adiabatic components are typically much longer than their counterparts. The long device lengths ensure that the mode evolution is slow enough to avoid radiation losses. Using small minimum feature sizes could reduce the device length, which has been demonstrated in the case of a 1 × 2 adiabatic power splitter with a 30 nm minimum feature size, resulting in a coupling length of 5 μm [39]. Unfortunately, such small features are well beyond the fabrication limitations set by foundries. The majority of the adiabatic power splitters with larger minimum feature sizes have linear taper profiles [40–46], whereas the linear taper profiles are not ideal for adiabatic transitions, and using non-linear profiles can significantly reduce the device lengths even with large minimum feature sizes.

Methods such as shortcuts to adiabaticity [47,48] and quasi-adiabatic dynamics [49–51] have been used to design adiabatic taper profiles that have resulted in 2 × 2 3 dB couplers with much shorter lengths as compared to the devices with linear taper profiles.

Reference [51] demonstrated over an order-of-magnitude reduction in the length of a 2 × 2 3 dB coupler as compared to a 3 dB coupler with a linear taper profile, where a 11.7 μm device length and a 3 ± 0.5 dB bandwidth of 75 nm were achieved. Other methods for designing adiabatic power splitters include ensuring that the adiabatic transition losses are maintained below a certain level by engineering the rate of change of the taper width profile [52]. In Reference [53], we developed a different approach, where the width profiles of the adiabatic tapers were expressed in terms of a polynomial function and the coefficients of the polynomial terms were optimized using numerical methods for the lowest losses for a given taper length. This method has the advantage of providing more design flexibility for the taper. Using this approach, we showed a reduction by half in the length of an adiabatic fiber-to-chip light coupler with no additional losses. More recently, we also applied this technique to a 1 × 2 Y-junction power splitter with a 120 nm minimum feature size and showed less than 0.25 dB and 0.23 dB losses for the TE and TM modes, respectively, in the spectral range of 1500 nm and 1600 nm [54]. The polynomial taper optimization is yet to be applied for the design of adiabatic power splitters with large minimum feature sizes, and the performances of these devices are to be analyzed when a CMOS-compatible fabrication process is used.

In this work, we propose and implemented a polynomial-based adiabatic taper profile optimization algorithm for the design of 1 × 2 and 2 × 2 adiabatic optical power splitters. The designed 1 × 2 Y-junction power splitters yielded coupling lengths ranging from 10 μm to 30 μm for minimum feature sizes ranging from 140 nm to 200 nm. The experimental results showed that a device with a 180 nm minimum feature size can have a coupling length of 20 μm with losses lower than 0.5 dB for both the TE and TM modes between 1480 nm and 1585 nm, where the splitting ratio was measured to be within 50 ± 2%. Besides, the 1 × 2 Y-junction architecture can be modified to allow arbitrary power splitting ratios by tuning the gap between the coupling waveguides. We designed power splitters with ratios of 50:50, 58:42, 68:32, 77:23, and 89:11. The measured standard deviations in the power splitting ratios for these 1 × 2 Y-junction splitters were between two to six percent. In addition, a broadband 2 × 2 3 dB power splitter with a 16 μm coupling region was measured to have 0.7 dB losses and splitting ratios within 50 ± 3% over a bandwidth of 130 nm, where a larger bandwidth measurement was not possible due to the range of our tunable laser source. Our experiments were performed over nine different chips, and the results indicated the excellent consistency of the performance and significantly fabrication-tolerant power splitters.

## 2. The 1 × 2 Y-Junction Optical Power Splitters

In this section, we applied our polynomial-based taper profile optimization algorithm to the design of adiabatic 1 × 2 Y-junction power splitters for symmetric power splitting (50:50) and arbitrary ratio power splitting. The optimization algorithm and the design procedure are defined, followed by the experimental results.

### 2.1. Design Methodology for 1 × 2 Y-Junction Power Splitters

A schematic of the 1 × 2 Y-junction power splitters is shown in Figure 1a. The width of the tips is denoted by $w_{tip}$, and the gaps are denoted by $w_{gap,1}$ and $w_{gap,2}$. The first stage of the Y-junction splitter is an adiabatically tapered region that gradually splits the input power from the middle waveguide to the upper and lower waveguides. The middle waveguide varies from 500 nm to $w_{tip}$, while the upper and lower waveguides vary from $w_{tip}$ to 500 nm over the taper length ($L_Y$). The value of $w_{tip}$ is the smallest feature in this design. A smaller $w_{tip}$ results in shorter devices, but it is typically limited by the minimum feature that is allowed by the foundry. The values of $w_{gap,1}$ and $w_{gap,2}$ are the widths of the lower and uppers gaps, respectively, and they are always chosen to be at least 20 nm wider than $w_{tip}$ to reduce potential reflections due to the abrupt changes at the two ends of the tapers. After the adiabatic coupling region, the waveguides are separated by 4 μm by using

two 15 µm-long S-bends, whose lengths were chosen generously to ensure the losses are predominantly due to the coupling region.

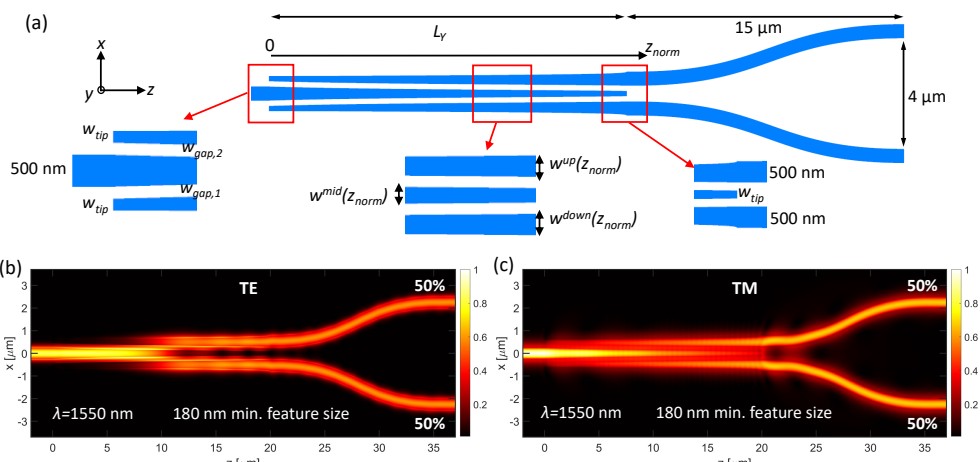

**Figure 1.** (**a**) Schematic of the 1 × 2 Y-junction power splitters. Electric fields along the Y-junctions for the (**b**) TE and (**c**) TM modes at a 1550 nm wavelength.

When $w_{gap,1}$ and $w_{gap,2}$ are equal, the Y-junction power splitter has a 50:50 splitting ratio. An imbalance between $w_{gap,1}$ and $w_{gap,2}$ causes the splitting ratio to vary. Here, we designed four devices with 50:50 splitting ratios with minimum feature sizes (i.e., $w_{tip}$) of 140 nm, 160 nm, 180 nm, and 200 nm. The designs with different $w_{tip}$ allowed us to observe the effect of the minimum feature size on the performance of the Y-junction power splitters. For these devices, $w_{gap,1}$ and $w_{gap,2}$ were equal, and they were 20 nm wider than $w_{tip}$. To realize Y-junctions with arbitrary splitting ratios, we kept $w_{tip}$ and $w_{gap,1}$ fixed at 180 nm and 200 nm, respectively, and varied $w_{gap,2}$ depending on the power splitting ratio we wished to achieve.

The width of the upper and lower tapers is defined by a polynomial function with eight terms, given by

$$w^{up}(z_{norm}) = \sum_{n=1}^{8} c_n z_{norm}^{p_n}, \tag{1}$$

where $p_n$ is the power of the $n$th term and $c_n$ is the coefficient of the $n$th term. The chosen $p$-values were [0, 0.3, 0.5, 0.7, 1, 2, 5, 10], and $c_n$ values were to be determined by the optimization. The normalized length ($z_{norm}$) is defined as $z_{norm} = z/L_Y$ and was limited by the bounds of [0, 1]. Using the polynomial function in Equation (1) allowed great freedom in modifying the shape of the taper. The selection of the $p$-values was made by considering how rapidly or gradually the tapers were desired to vary in width along the $z$-axis. For example, for a $p$-value of 0.3, the taper became wider quickly, then the width changed slowly. On the other hand, for a $p$-value of 10, the shape of the taper changed very slowly until the end of the taper, where the width started changing rapidly. We chose the $p$-values of 0.3 and 10 by trial, where we ensured that the taper shape did not vary too quickly, hence disturbing the adiabatic evolution at the beginning or at the end of the taper. Then, we added five more $p$-values between 0.3 and 10 to ensure that the taper could take shape freely between the two extremes. To satisfy the boundary conditions (i.e., $w^{up}(0) = w_{tip}$ and $w^{up}(1) = 500$ nm), we ensured that

$$c_1 = w_{tip} \ \& \ p_1 = 0 \ \& \ \sum_{n=1}^{8} c_n = 500 \text{ nm.} \tag{2}$$

We define the width of the middle taper in terms of the widths of the upper and lower tapers (i.e., $w^{up}$):

$$w^{mid}(z_{norm}) = 500 \text{ nm} - (500 \text{ nm} - w_{tip}) \cdot \left( \frac{w^{up}(z_{norm}) - w_{tip}}{500 \text{ nm} - w_{tip}} \right)^{\alpha}, \tag{3}$$

where $\alpha$ is a parameter that determines the relation between the widths of the upper and middle tapers. $\alpha$ served as a value that provides an additional degree of freedom in the design and caused the profile of the middle taper to vary differently from the profiles of the upper and lower tapers.

The Y-junction power splitters were modeled with the Lumerical Eigenmode Expansion (EME) solver. With the EME method, the 3D device was divided into 2D slices, where the finite-difference method was used to calculate the eigenmodes at each slice. Once the eigenmodes were calculated, mode evolution could be calculated along the taper with great accuracy. A significant advantage of the EME method is that the distance between the cells can be modified after the calculation of the eigenmodes to generate the desired taper profiles without the need for recalculating the eigenmodes. The $p_n$ values were chosen prior to the optimization, and the taper profiles were optimized by optimizing the $c_n$ values. A wide range of $p$-values was chosen to allow for a wide range of taper shapes. Small $p$-values ($p < 1$) allowed for a rapid change of shape at the beginning of the taper, followed by a more-gradual change towards the end of the taper, while larger $p$-values ($p > 1$) caused the taper shape to vary slowly, followed by a rapid change of width. The $c_n$ values determined to what degree these shapes should be included in the optimized structure. A particle swarm optimization algorithm was used to optimize the $c_n$ values. The optimization goal was set to minimize the taper losses at the wavelength of 1550 nm for the TE, or the TM mode, whichever was higher, so as to achieve polarization-insensitive operation. The number of particles was selected to be 45, and the optimization was run for 25 generations, or until the result had not improved for 3 consecutive generations, whichever happened first. After the optimization, we simulated the devices with the -D finite-difference time-domain (FDTD) method to ensure the accuracy of the results. The field profiles along an optimized $1 \times 2$ 50:50 Y-junction power splitter for a minimum feature size of 180 nm (i.e., $w_{tip} = 180$ nm, $w_{gap,1} = w_{gap,2} = 200$ nm) are shown for the TE and TM modes in Figure 1b,c, respectively.

### 2.2. Profile-optimized $1 \times 2$ Y-Junction Power Splitters

To understand the effect of the $\alpha$ parameter on the taper losses, we optimized a 50:50 Y-junction power splitter with a 180 nm minimum feature size for $\alpha$ values ranging from 0.2 to 1. The simulated taper losses for the TE mode, which is typically the lossier mode for Y-junction power splitters [55], as a function of the Y-junction length, are presented in Figure 2a. For all the $\alpha$ values we analyzed here, the required Y-junction lengths were drastically shorter than a linear Y-junction, which is shown with the dashed lines. We note that, for an $\alpha$ value of 0.5 and lower, the taper losses were minimized rapidly. Though there was little difference in the simulated taper losses' $\alpha$ values between 0.2 and 0.5, we preferred to select an $\alpha$ of 0.5 because very small or very large values of $\alpha$ could cause the middle taper to vary too rapidly either at the beginning or the end of the taper, which could make the taper more susceptible to fabrication errors. Moreover, we observed that an $\alpha$ value of 0.5 worked well for all other minimum feature sizes, and hence, we continued our optimizations by setting $\alpha$ equal to 0.5.

Figure 2b,c show the minimized taper losses for the TE and TM modes, respectively, at a wavelength of 1550 nm and for an $\alpha$ parameter of 0.5. For both the TE and TM modes, devices with smaller minimum feature sizes tended to require shorter Y-junction lengths to achieve low losses ($< \sim 0.05$ dB). To ensure that the taper losses were minimal for both modes, we chose coupling lengths of 10 μm, 15 μm, 20 μm, and 30 μm for the minimum feature sizes of 140 nm, 160 nm, 180 nm, and 200 nm, respectively.

The excess losses for the optimized devices were simulated with the FDTD method and are shown in Figure 2d for the four minimum feature sizes. For the TE mode, all minimum feature sizes achieved very low excess losses around 1550 nm, as expected from the losses calculated with the EME solver. However, at shorter wavelengths, the excess losses increased, which was more pronounced for the TE mode and for larger minimum feature sizes. This was because, at shorter wavelengths, the TE mode was more strongly confined in the Si core; hence, the Y-junction length needed to be longer as compared to

longer wavelengths for low-loss power splitting. Yet, the excess losses for the TE mode were lower than 0.2 dB for wavelengths longer than 1500 nm. For the TM mode, on the other hand, the excess losses were much less wavelength-dependent and were between 0.15 dB and 0.25 dB in the whole spectrum range. These values were slightly above the taper losses estimated by the EME calculations. This discrepancy was because of the abrupt changes at the beginning and end of the tapers, which caused an extra ∼0.1 dB loss, which was not included in the EME analysis. Our further analysis showed that, for a variation of ±20 nm in the widths of the waveguides, the calculated losses varied by less than 0.1 dB within the wavelength range of 1450–1650 nm, indicating that the 1 × 2 Y-junction power splitters could tolerate a large amount of fabrication errors.

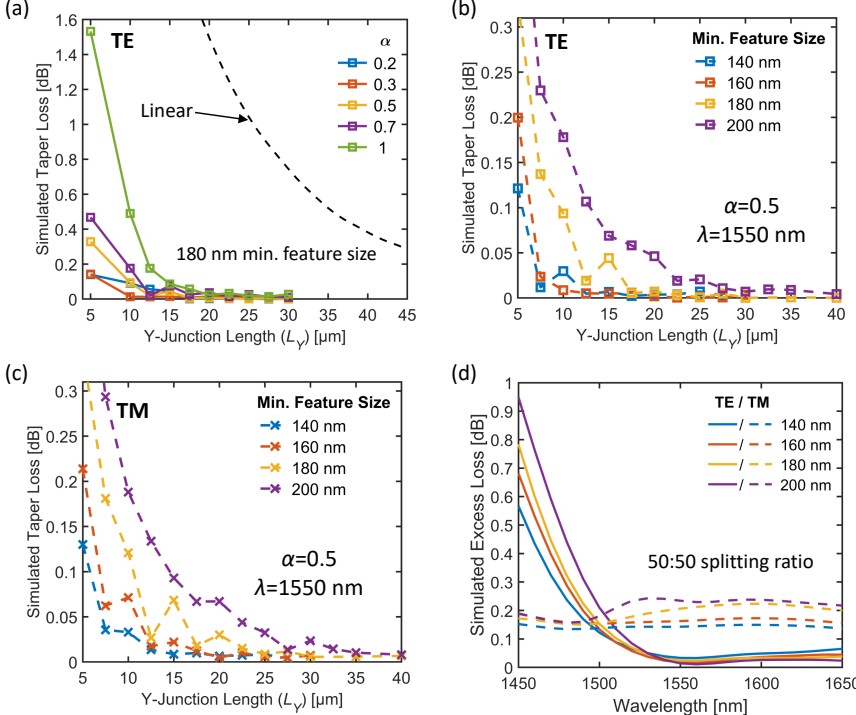

**Figure 2.** (**a**) Simulated taper losses for the Y-junction power splitter with a 180 nm minimum feature size for varying $\alpha$ for the TE mode. The optimized taper losses for the (**b**) TE and (**c**) TM modes. (**d**) Spectra of the excess losses for the devices with four minimum feature sizes.

To design 1 × 2 Y-junctions with arbitrary power splitting ratios, we chose a minimum feature size of 180 nm and kept $w_{tip}$ and $w_{gap,1}$ fixed at 180 nm and 200 nm, respectively, while varying $w_{gap,2}$. For each value of $w_{gap,2}$, we optimized the profile of the device to minimize the taper losses for the TE mode. Although arbitrary splitting ratios were achievable for both modes, for a given $w_{gap,2}$ value, Y-junctions will have different splitting ratios for the TE and TM modes. To prove the effectiveness of this structure, we designed our devices for the TE mode only, although the same optimization can be performed for the TM mode. The ratio of the transmitted power to the lower arm of the Y-junction power splitter is shown in Figure 3a for the TE mode as a function $w_{gap,2}$. Power splitting ratios from 50:50 to 98:2 could be achieved by varying the $w_{gap,2}$ from 200 nm to 520 nm.

We chose values for $w_{gap,2}$ of 200 nm, 220 nm, 250 nm, 290 nm, and 355 nm, which resulted in power splitting ratios of 50:50, 58:42, 68:32, 77:23, and 89:11, respectively. The spectra of the power ratios at the lower arm are shown in Figure 3b. The spectral variations in the splitting ratios were within ±1% for all the Y-junction power splitters throughout the wavelength range of 1450 nm to 1650 nm. The excess losses for these devices are shown in Figure 3c, where the excess losses were below 0.2 dB for wavelengths longer than 1500 nm. The electric field profiles (|E|) along the Y-junction splitters showed that the power splitting took place with little radiation losses at a 1550 nm wavelength for a

$w_{gap,2}$ of 250 nm and 355, as shown in Figure 3d,e, respectively. The geometrical parameters of the designed devices are presented in Table A1.

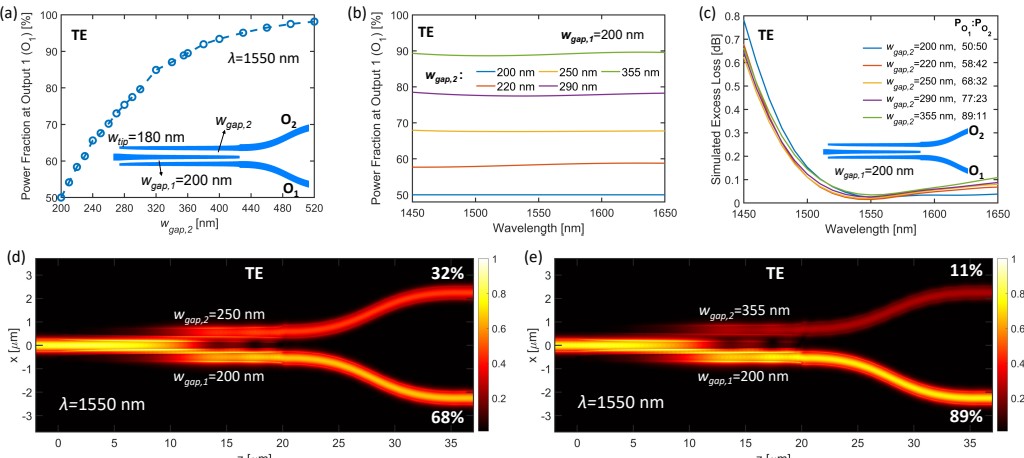

**Figure 3.** (**a**) Power fraction at the lower arm for varying $w_{gap,2}$. (**b**) Spectra of the splitting ratios for the Y-junction power splitters for varying $w_{gap,2}$. (**c**) Spectra of the excess losses for the devices with arbitrary power splitting ratios. The magnitude of electric fields ($|E|$) along the Y-junction power splitter with (**d**) $w_{gap,2} = 250$ nm and (**e**) $w_{gap,2} = 355$ nm, at 1550 nm. All figures pertain to the TE mode.

### 2.3. Experimental Results for 1 × 2 Y-Junction Power Splitters

The 1 × 2 Y-junction power splitters were fabricated at the Advanced Micro Foundry (AMF), in Singapore, on a 220 nm SOI platform with a 3 µm buried oxide layer. The devices were then coated with a 2.3 µm-thick $SiO_2$ cladding layer. The edges of the chips were deeply etched to allow access to the edge couplers. Scanning electron microscope images of the resultant Y-junctions with 50:50 and 89:11 power splitting ratios are shown in Figure 4a,b, respectively, for a minimum feature size of 180 nm.

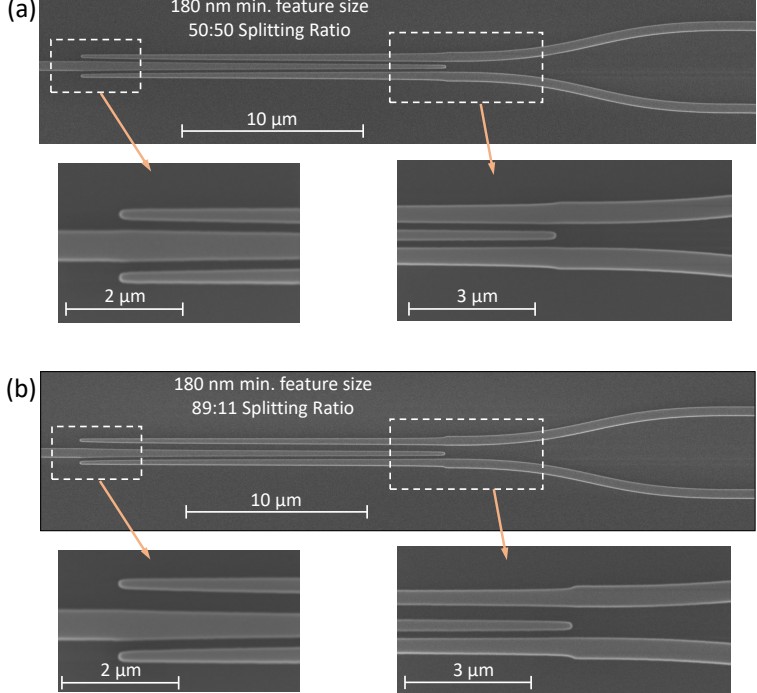

**Figure 4.** Scanning electron microscope images of the (**a**) 50:50 power splitter and (**b**) 89:11 power splitter with 180 nm minimum feature sizes.

To measure the excess losses, light from a tunable laser was passed through a waveplate and a linear polarizer to control the polarization of the light. Ten percent of the light was then sampled using a fiber-based power splitter in order to measure the power of the light before the chip. The rest of the light was coupled into the chip. Light coupling into and out of the chip was performed with polarization maintaining lensed fibers with a 5 µm spot size. The lensed fiber at the output was connected to a fiber-based polarization beam splitter, to ensure the measured output power belonged to either the TE or the TM mode for all of nine chips, and any depolarization that might have occurred in the chip or in the fibers was not measured. The transmission was obtained from the ratio of the output power to the sampled input power and was normalized with a reference measurement, whose path consisted of two edge couplers and a waveguide connecting the edge couplers. Transmission through the Y-junction power splitters was then measured, and the excess losses were calculated after measuring the transmission through both arms of the devices. The excess losses were measured on nine different chips, and the results were averaged and the standard deviations calculated. The standard deviation in the transmission due to the variations of the losses of the edge couplers and waveguides were measured to be 0.1 dB for the TE mode and 0.2 dB for the TM mode for the same chip, with nearly no wavelength sensitivity. For measurements involving Mach–Zehnder interferometers (MZIs), we used a broadband superluminescent diode source centered at 1550 nm with a 3 dB bandwidth of 50 nm and measured the output power with an optical spectrum analyzer. The MZI spectra were then used to deduce the extinction ratios of the peaks and dips, which was used as a metric for how equally a 50:50 splitter splits the optical power. A good extinction ratio is typically over 20 dB, which ensures that the power splitting ratio is confined within 50 ± 5%.

The average measured excess losses for the 50:50 Y-junction power splitters are shown in Figure 5a (TE) and Figure 5b (TM) for the four different minimum feature sizes. The Y-junction power splitters with 140 nm and 160 nm minimum feature sizes performed poorly, as was evident from the excess losses higher than 1 dB. The devices with larger minimum feature sizes (i.e., 180 nm and 200 nm), on the other hand, showed drastically lower losses. Although features as small as 140 nm are allowed by the foundry, the significant minimum feature size dependency showed that the fabrication process is better suited for devices with larger features. Among the four minimum feature sizes, the 180 nm minimum feature size set had losses lower than 0.4 dB for both the TE and TM modes in the wavelength range of 1490 nm and 1590 nm, proving that polarization-insensitive operation was realized with the profile-optimized Y-junctions.

Figure 5c,d show the average excess losses for the TE and TM modes, respectively, for a minimum feature size of 180 nm. The shaded regions show one standard deviation above and below the average excess losses, calculated from our measurements on nine chips. The standard deviation for the TE and TM modes were relatively constant around 0.2 and 0.3 dB, respectively, throughout the spectrum. These values were comparable to the deviations among different edge couplers on the same chip, which were 0.1 dB and 0.2 dB for the TE and TM modes, respectively. Yet, it is important to note that the standard deviations Figure 5c,d are a combination of deviations in light coupling losses, waveguide losses, and the losses of the Y-junction power splitters.

To analyze the balance in power splitting, an MZI with a length imbalance of 204 µm was constructed with two Y-junction power splitters. Representative MZI transmission spectra are shown in Figure 5e,f for the TE and TM modes, respectively, for the 180 nm minimum feature sizes. The extinction ratios between the peaks and the dips were over 20 dB for the wavelength range of 1460 nm to 1610 nm for both modes. When the average of the extinction ratios was concerned, the 180 nm minimum feature size outperformed the other minimum feature sizes, as shown in Figure 5g,h. The average extinction ratios were over 24 dB between 1480 nm and 1590 nm for both modes for the 180 nm minimum feature size. Although Y-junctions with 160 nm and 200 nm minimum feature sizes had 20 dB extinction ratios over a broadband for both modes, the Y-junction with a 140 nm

minimum feature size performed poorly with an extinction ratio under 20 dB for the TE mode. It is worthwhile to note that we were unable to extract the extinction ratios from some of the MZI spectra for Y-junctions with 140 nm and 160 nm minimum feature sizes due to low extinction ratios or a noisy spectrum. Hence, the extinction ratios for devices with 140 nm and 160 nm minimum feature sizes in Figure 5g,h were averaged over four and five spectra, respectively. For the devices with 180 nm and 200 nm minimum feature sizes, the average extinction ratios were calculated from all nine devices we tested. We previously observed a similar minimum feature size dependency of the splitting ratios, where devices with moderate minimum feature sizes performed better in terms of loss and splitting ratio than devices with small and large minimum feature sizes [55].

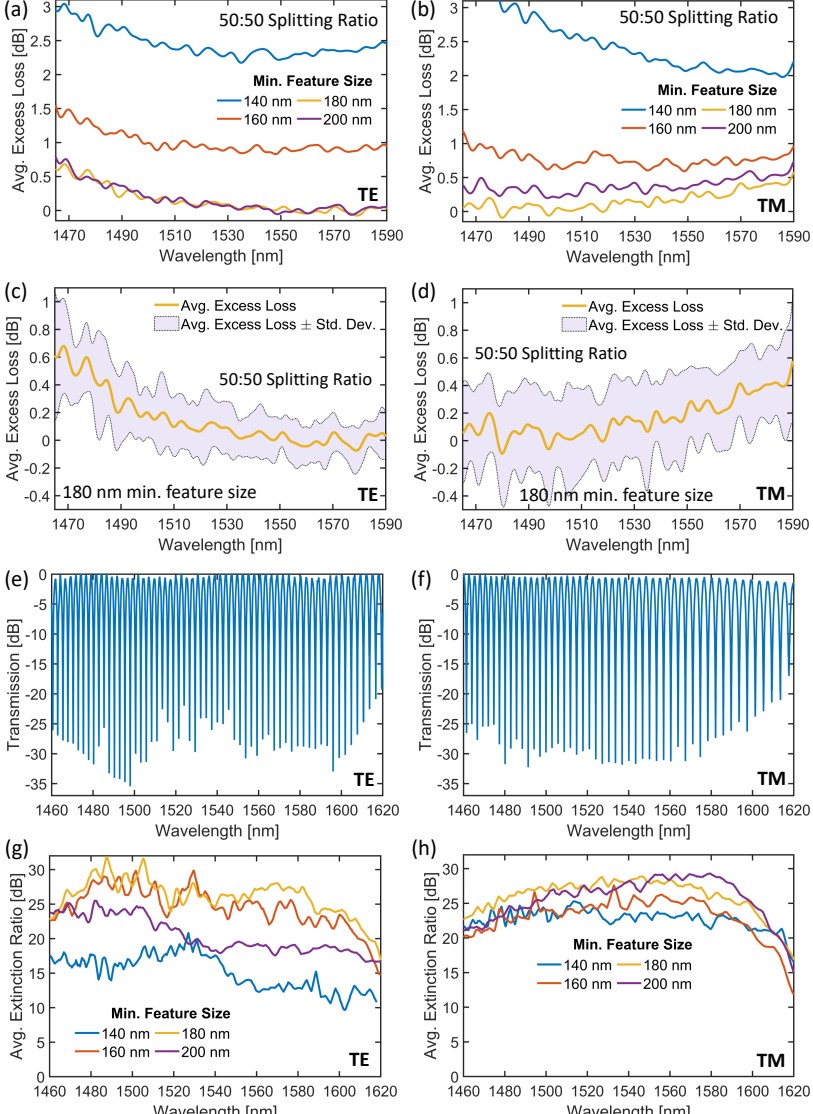

**Figure 5.** Average excess losses for the 50:50 1 × 2 Y-junction for the (**a**) TE and (**b**) TM modes. Average excess losses and standard deviation in losses for the Y-junction with 180 nm minimum feature size for the (**c**) TE and (**d**) TM modes. Transmission spectra of the Mach–Zehnder interferometer for the Y-junction with a 180 nm minimum feature size for the (**e**) TE and (**f**) TM modes. Spectra of the average extinction ratios for the (**g**) TE and (**h**) TM modes.

The average excess losses for the TE mode of the arbitrary ratio power splitters are shown in Figure 6a for varying $w_{gap,2}$. All the devices we analyzed here had excess losses lower than 0.6 dB for wavelengths longer than 1490 nm, indicating that the varying of the splitting ratio did not affect the bandwidth significantly. Figure 6b shows the spectra of

the average power ratio at the lower arm of the Y-junction power splitter along with the simulated results shown as dashed lines. For the power splitters with designed splitting ratios of 50:50, and 58:42, the experimental results were in excellent agreement with the simulated splitting ratios. For the splitters with designed splitting ratios of 68:32, 77:23, and 89:11, however, the differences were 3%, 7%, and 5%, respectively, in the C-band. This discrepancy might have been caused by the gaps between the waveguides being affected disproportionately by the fabrication errors. Such fabrication errors typically do not affect the results for splitting ratios close to 50:50, thanks to the inherent fabrication tolerance of the adiabatic power splitters, but they are more pronounced when the asymmetry in the structure is higher.

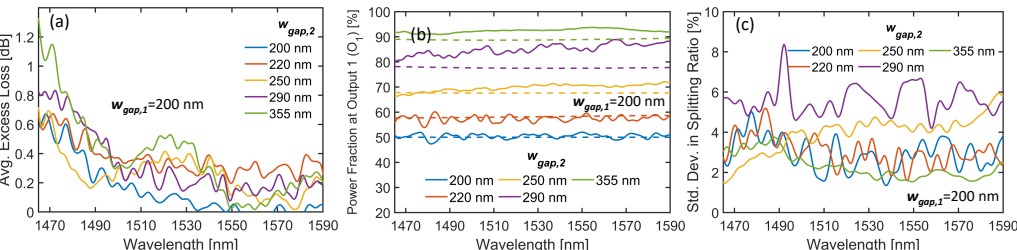

**Figure 6.** (**a**) Average excess losses for the 1 × 2 Y-junction with arbitrary splitting ratios. (**b**) The average power ratio at the lower arm of the Y-junction for the power splitters with different splitting ratios (solid lines) and their corresponding simulation results (dashed lines). (**c**) Standard deviation in the measured splitting ratios of the 1 × 2 Y-junction power splitters.

The standard deviations in the splitting ratios on different chips are shown in Figure 6c for the varying $w_{gap,2}$ values. The standard deviations ranged between 2% and 4% with the exception of a $w_{gap,2}$ of 290 nm, where the deviation was around 6%.

## 3. Broadband 2 × 2 50:50 Optical Power Splitters on 90 nm Slab

In this section, we present our 2 × 2 50:50 optical power splitters fabricated on a 90 nm Si slab along with the taper profile optimization. By doing so, we aimed to show that the proposed polynomial taper profile optimization can accommodate different types of power splitters with different architectures. We discuss the design procedure, experimental excess losses, and splitting ratios of the devices.

### 3.1. Design of 2 × 2 50:50 Power Splitters

A schematic of the 2 × 2 50:50 power splitter is shown in Figure 7a. The device architecture was borrowed from [46]. This power splitter was designed on a 90 nm Si slab, which is a commonly offered thickness by foundries. The 90 nm-thick slab causes the modes to leak from one waveguide to the other; therefore, the device shows a significant reduction in the coupling length as compared to a device with a linear coupling region. The downside of this approach is that the slab does not support the TM mode, hence this particular geometry only operates with the TE mode. However, the 2 × 2 50:50 power splitters can also be designed (and optimized) without the 90 nm slab and, hence, can be made to operate with both the TE and TM mode.

The use of a 90 nm Si slab requires a conversion from the bus-waveguide to rib-waveguide modes. Prior to the conversion, the upper input waveguide was tapered down to 400 nm, while the lower input waveguide was tapered up to 600 nm, over a 1 µm-long taper. Then, 3 µm-long profile-optimized tapers on the 90 nm slab were used, while the widths of the 220 nm Si ribs remained constant. This taper was followed by 25 µm-long Bezier bends that brought the gap between the waveguides from 4 µm down to 200 nm. Long bends are needed to ensure low losses, since rib-waveguides are typically lossy when sharp bends are used. Furthermore, when the waveguides become closer, the coupling between waveguide modes becomes stronger; hence, bends need to have a larger radius as the bends become closer to each other.

In the adiabatic coupling stage, the upper and lower waveguides were tapered from 400 nm and 600 nm, respectively, to a width of 500 nm. The mode on the upper arm was transformed into the anti-symmetric waveguide mode at the output of the adiabatic coupling stage, while the mode in the lower arm was transformed into the symmetric mode, whose profiles are shown in Figure 7a. The width of the upper waveguide is given by

$$w^{up}(z_{norm}) = \sum_{m=1}^{8} c_n z_{norm}^{p_n}. \tag{4}$$

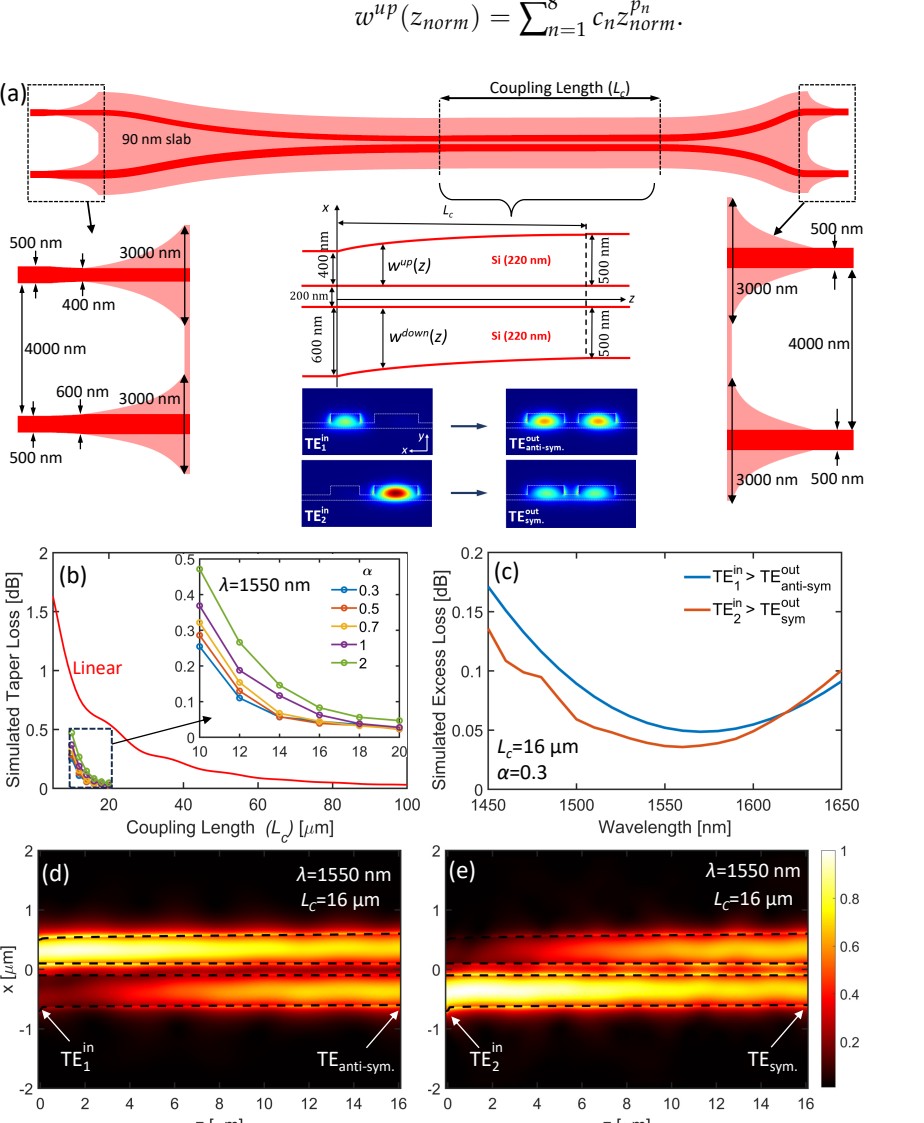

**Figure 7.** (**a**) Schematic of the designed 2 × 2 power splitter. (**b**) Simulated taper losses in the adiabatic coupling region. (**c**) Spectra of the simulated excess losses when power is launched from both inputs. The mode field profiles along the coupling region for (**d**) Input 1 (TE$_1$ mode) and (**e**) Input 2 (TE$_2$ mode).

The selected $p_n$ values were [0, 0.1, 0.3, 0.5, 0.7, 1, 3, 5]. These values were smaller than the ones for the 1 × 2 Y-junction power splitter. This was because the 2 × 2 50:50 power splitters performed better with smaller *p*-values, and including 0.1 in the *p*-values reduced the times to find the optimal structures. To satisfy the boundary conditions (i.e., $w^{up}(0)$ = 400 nm, and $w^{up}(1)$ = 500 nm), we ensured that

$$c_1 = 400 \text{ nm } \& \ p_1 = 0 \ \& \ \sum_{n=1}^{8} c_n = 500 \text{ nm}. \tag{5}$$

We define the width of the lower waveguide in terms of the width of the upper waveguide according to

$$w^{down}(z_{norm}) = 600 \text{ nm} - (600 \text{ nm} - 500 \text{ nm}) . \left( \frac{w^{up}(z_{norm}) - 500 \text{ nm}}{600 \text{ nm} - 500 \text{ nm}} \right)^{\alpha}, \tag{6}$$

where $\alpha$ is a parameter, similar to the one used in the optimization of the 1 × 2 Y-junction power splitter. Since two TE mode evolutions take place in the adiabatic coupling stage, we minimized whichever transition had higher losses. The adiabatic coupler stage was followed by 11 µm-long Bezier bends and a rib-waveguide to bus-waveguide converter with a length of 2 µm.

Figure 7b shows the optimized taper losses as a function of the coupler length ($L_c$), along with the linear taper (red solid line) for comparison. The linear taper required approximately a 60 µm coupling length to achieve losses lower than 0.1 dB. On the other hand, the profile-optimized tapers could achieve 0.1 dB losses with a coupling length of approximately 12 µm and 16 µm, depending on the $\alpha$ parameter. Among the $\alpha$ parameters analyzed here, an $\alpha$ of 0.3 achieved the lowest losses with the shortest length; hence, we used an $\alpha$ parameter of 0.3 and a coupling length of 16 µm. When the coupling region was simulated with the 3D FDTD method, an excess loss of less than 0.2 dB can be achieved over a bandwidth of 200 nm, as shown in Figure 7c. Moreover, the excess loss difference between the two inputs was less than 0.05 dB throughout the spectrum, indicating a balance in losses seen from both inputs. When we included fabrication errors of ±20 nm in the widths of the two waveguides, the calculated losses varied by less than 0.09 dB within the wavelength range of 1450–1650 nm. This indicated that the designed 2 × 2 Y-junction power splitter was tolerant to fabrication errors. Figure 7d,e show the field profiles of the mode evolution from the inputs to the anti-symmetric and symmetric modes, respectively. The mode profiles showed a successful transition from the inputs to the desired output modes. The geometrical parameters of the device is given in Table A1.

### 3.2. Experimental Design for the 2 × 2 50:50 Power Splitters

A scanning electron microscope image of the 2 × 2 50:50 power splitter is shown in Figure 8a. The testing for the 2 × 2 50:50 power splitters was performed similar to that of the 1 × 2 Y-junction power splitters. The transmission through the 2 × 2 50:50 power splitters was measured from both inputs to both outputs; hence, there were four measurements for a single device, whereas two measurements were taken for the 1 × 2 Y-junction power splitters. The excess losses were obtained for each input. The average excess losses of the 2 × 2 50:50 power splitter are shown in Figure 8b,c when the input power was launched from Input 1 and Input 2 of the device, respectively. The excess losses fluctuated around 0.7 dB, with a standard deviation of approximately ±0.3 dB. The excess losses demonstrated excellent wavelength insensitivity in the 1460 nm- to 1590 nm-wavelength range. The standard deviations for the 2 × 2 50:50 power splitter were higher when compared to the 1 × 2 Y-junction power splitter. This could be because the 2 × 2 50:50 power splitter had more stages, which caused more overall deviations. The mode conversion losses from the 500 nm-wide Si bus-waveguide to the waveguide with 90 nm slab made up some of the excess losses and deviations in the loss. Lower excess losses could be achieved if this power splitter were used along with other devices that are already on a slab, such as programmable optical meshes [3].

Transmission through MZIs, consisting of two devices connected in series with a length imbalance of 160 160 µm, was measured from both inputs and both outputs, totaling four measurements per interferometer—unlike the 1 × 2 Y-junction power splitters, which required only one measurement per interferometer. For this measurement, one of the 2 × 2 50:50 power splitters was rotated by 180 along the *z*-axis before creating the interferometer, so that if there were an imbalance in power splitting, it would be emphasized in the output. A representative transmission spectrum of an MZI is shown in Figure 8d. In the wavelength range of 1450 nm to 1610 nm, the extinction ratios when light power

was launched from both inputs were above 20 dB. The average extinction ratios, as seen in Figure 8e,f when the light was launched from Input 1 and Input 2, respectively, indicated extinction ratios greater than 20 dB for 1460 nm and 1595 nm. Hence, the designed power splitter had an excellent balance in the splitting ratio over a wide bandwidth. The data shown in Figure 8d–f were taken from Output 1 of the device, since the extinction ratios were observed to be nearly identical regardless of which output the light was collected from.

We summarize the key performance metrics for the selected devices in Table 1. Although we performed our experiments at room temperature, the deviation in the performance was not expected to be significant if the temperature was varied for adiabatic power splitters. For example, for an adiabatic 2 × 2 power splitter designed with fast quasi-adiabatic dynamics, the temperature was varied between 20 and 50 °C and the transmission outputs remained within the range of 3 ± 1 dB for both output arms at 1550 nm [50]. In addition, thermal simulations of a multi-mode interferometer-based power splitter showed that the transmission variations were less than 0.35% in the temperature range of 200–360 K [14].

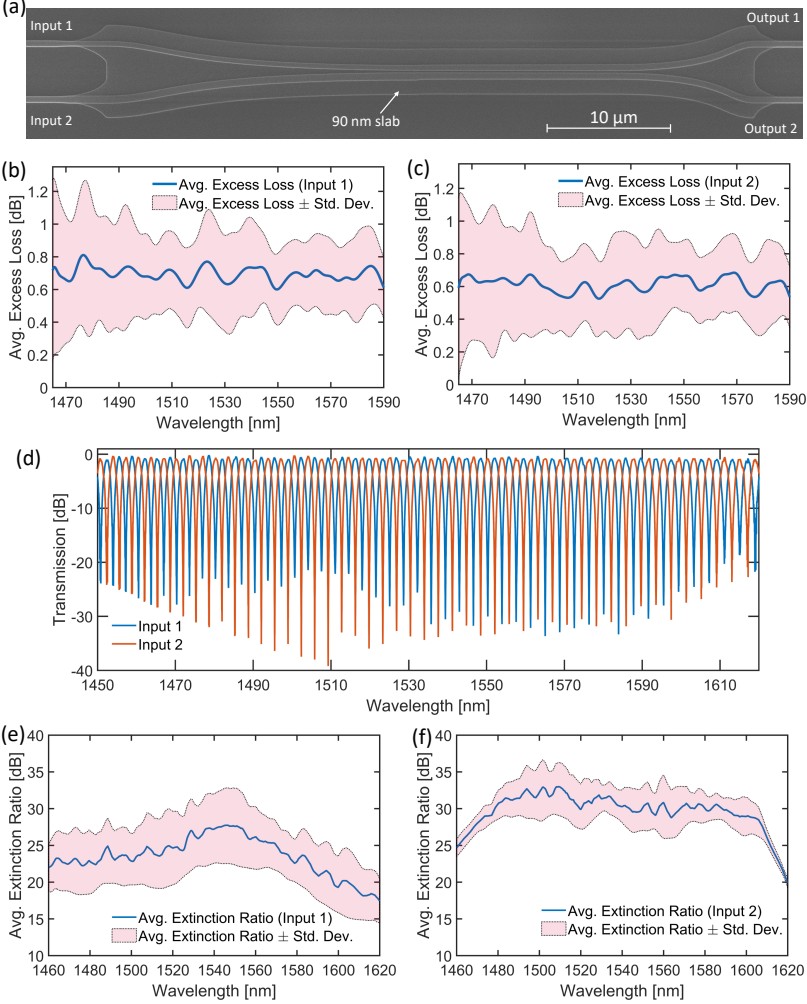

**Figure 8.** (**a**) Scanning electron microscope image of the 2 × 2 3 dB power splitter. Average excess losses and standard deviation in losses when the input power is launched from (**b**) Input 1 and (**c**) Input 2. (**d**) A representative transmission spectra of the Mach–Zehnder interferometer. Spectra of the average extinction ratios when the input power is launched from (**e**) Input 1 and (**f**) Input 2.

We compare the performance of our devices with the adiabatic devices in the literature in Table 2. Our 1 × 2 Y-junction power splitter demonstrated a short coupling length of 20 µm for a minimum feature size of 180 nm. Unless a smaller minimum feature size is used [39,55] or a Si slab is used [44], our 1 × 2 Y-junction power splitter achieved the

shortest coupling length (20 μm) for a minimum feature size of 180 nm and the largest bandwidth. Similarly, our optimized 2 × 2 power splitter showed a considerable reduction in the coupling length compared to devices with a minimum feature size of 200 nm and a broad bandwidth.

**Table 1.** Summary of measured performance metrics of the selected power splitters designed in this paper. The excess losses (ELs) and splitting ratios (SRs) are given for the entire bandwidth.

| Device | Min. Feature Size (nm) | $EL_{TE}$ [1] (dB) | $EL_{TM}$ [1] (dB) | $SR_{TE}$ [2] | Bandwidth (nm) |
|---|---|---|---|---|---|
| 1 × 2 50:50 Y-junction | 180 | <0.5 | <0.5 | 50 ± 2 | 1480–1585 |
| 1 × 2 58:42 Y-junction | 180 | <0.6 | - | 58 ± 2 | 1490–1590 |
| 1 × 2 68:32 Y-junction | 180 | <0.5 | - | 71 ± 3 | 1487–1590 |
| 1 × 2 78:22 Y-junction | 180 | <0.6 | - | 84 ± 5 | 1490–1590 |
| 1 × 2 89:11 Y-junction | 180 | <0.6 | - | 94 ± 2 | 1490–1590 |
| 2 × 2 50:50 | 200 | <0.7 | - | 50 ± 3 | 1465–1590 |

[1] EL: excess loss. [2] SR: splitting ratio.

**Table 2.** A comparison of the adiabatic 50:50 power splitters reported in the literature with our power splitters.

| Ref. | Device Type | Length (μm) | Min. Feature Size (nm) | $EL_{TE}$ [1] (dB) | $EL_{TM}$ [1] (dB) | Bandwidth (nm) |
|---|---|---|---|---|---|---|
| [39] | 1 × 2 | 5 | 30 | 0.19 | 0.14 | 1530–1600 |
| [41] | 1 × 2 | 40 | 100 | <1 | <1 | 1260–1650 |
| [44] | 1 × 2 | 19 | 200 | ∼0.09 | - | 1510–1560 |
| [45] | 1 × 2 | 40 | 200 | ∼0.06 | - | 1470–1570 |
| [55] | 1 × 2 | 14 | 120 | 0.25 | 0.23 | 1500–1600 |
| This work | 1 × 2 | 20 | 180 | <0.5 | <0.5 | 1480–1585 |
| [46] | 2 × 2 | 100 | 200 | NR [2] | - | 1500–1600 [3] |
| [47] | 2 × 2 | 60 | 200 | <0.3 | - | 1500–1600 |
| [48] | 2 × 2 | 83 | 200 | <0.15 | - | 1480–1620 |
| [49] | 2 × 2 | 26.3 | 200 | NR [2] | - | 1470–1620 [4] |
| [50] | 2 × 2 | 11.7 | 150 | NR [2] | - | 1500–1600 [4] |
| [51] | 2 × 2 | 11.7 | 150 | NR [2] | NR [2] | 1490–1565 [4] |
| This work | 2 × 2 | 16 | 200 | <0.7 | - | 1465–1590 |

[1] EL: Excess loss. [2] NR: not reported. [3] This is the measurement bandwidth. [4] These indicate a 3 ± 0.5 dB bandwidth.

## 4. Conclusions

In conclusion, we developed an adiabatic taper profile optimization algorithm for the design of 1 × 2 and 2 × 2 power splitters with low losses, a broad bandwidth, balanced splitting ratios, and compact sizes. The 1 × 2 Y-junction power splitters with a 180 nm minimum feature size and 20 μm coupling length showed average excess losses lower than 0.5 dB for both modes in the wavelength range of 1480 nm to 1585 nm. When the 1 × 2 Y-junction power splitter geometry was modified to achieve arbitrary splitting ratios, the excess losses were maintained below 0.6 dB across the same wavelength range, while achieving splitting ratios ranging from 50% to 94% with standard deviations ranging between 2% and 6%. Similarly, the 2 × 2 50:50 power splitters with a 16 μm coupling length were observed to have an average excess loss of 0.7 dB. Our device designs demonstrated excellent consistency across multiple chips, while ensuring low losses and a wide bandwidth. With the aforementioned power splitters, we demonstrated the effectiveness of a polynomial-based optimization algorithm for the minimization of the losses of adiabatic power splitters, which resulted in low-loss and highly fabrication-tolerant devices with significantly improved footprints, as compared to their adiabatic counterparts. This optimization is applicable to many other adiabatic components and can accommodate devices with a wide range of minimum feature sizes.

**Author Contributions:** Conceptualization, C.O.; methodology, C.O.; software, C.O.; validation, C.O.; formal analysis, C.O.; investigation, C.O.; writing—original draft preparation, C.O., J.S.A. and M.M.; writing—review and editing, C.O., J.S.A. and M.M.; visualization, C.O.; supervision, J.S.A. and M.M. All authors have read and agreed to the published version of the manuscript.

**Funding:** This research received no external funding.

**Institutional Review Board Statement:** Not applicable.

**Informed Consent Statement:** Not applicable.

**Data Availability Statement:** The data presented in this study are available upon request from the corresponding author.

**Acknowledgments:** The authors would like to thank CMC Microsystems for arranging the fabrication run at the Advanced Micro Foundry (AMF), Singapore.

**Conflicts of Interest:** The authors declare no conflict of interest.

## Appendix A

Table A1 shows the optimized structural parameters of the optimized devices presented throughout the paper.

**Table A1.** Optimized parameters of the power splitters.

| Device | Min. Feature Size (nm) | Coupling Length (μm) | $\alpha$ | $p_n$ | $c_n$ |
|---|---|---|---|---|---|
| 1 × 2 50:50 Y-junction | 140 | 10 | 0.5 | [0, 0.3, 0.5, 0.7, 1, 2, 5, 10] | [140, 0, 0, 0, 170, 37, 153] |
| | 160 | 15 | | | [160, 0, 0, 50, 68, 127, 11, 84] |
| | 180 | 20 | | | [180, 0, 1, 117, 52, 57, 82, 11] |
| | 200 | 30 | | | [200, 0, 7, 73, 71, 33, 16, 100] |
| 1 × 2 58:42 Y-junction | 180 | 20 | 0.5 | [0, 0.3, 0.5, 0.7, 1, 2, 5, 10] | [180, 22, 49, 57, 8, 56, 47, 81] |
| 1 × 2 68:32 Y-junction | 180 | 20 | | | [180, 35, 57, 78, 5, 31, 54, 60] |
| 1 × 2 78:22 Y-junction | 180 | 20 | | | [180, 60, 43, 40, 33, 21, 57, 66] |
| 1 × 2 89:11 Y-junction | 180 | 20 | | | [180, 60, 79, 9, 13, 16, 56, 87] |
| 2 × 2 50:50 | 200 | 16 | 0.3 | [0, 0.1, 0.3, 0.5, 0.7, 1, 3, 5] | [400, 14, 45, 18, 1, 10, 0, 12] |

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
