# Peer review of "Foundry-Processed Compact and Broadband Adiabatic Optical Power Splitters with Strong Fabrication Tolerance"

_photonics, doi:10.3390/photonics10121310_

Round 1

Reviewer 1 Report

Comments and Suggestions for Authors

see the attachment.

Reviewer 2 Report

Comments and Suggestions for Authors

In this manuscript, the authors used a polynomial taper optimization algorithm to design and process adiabatic power splitters with shorter length than other adiabatic power splitters. The authors also provide a detailed review of the progress of optical power splitters. This work is complete and interesting, which makes sense for improving the integration of optical chips. However, there are certain aspects of the content that might need clarification, and I would like to recommend publication after the authors have considered the following questions:

1. The values of some parameters are confusing. For example, in Equations 1 & 4, the value of p is a sequence whose meaning is not clearly stated. What are these sequences generated based on? What do the starting and ending values of this sequence mean respectively? In the Equation 3, the author states that for the α value of 0.5 and lower, the taper losses are minimized rapidly. According to Fig.2(a), smaller values result in better performance. So why not choose a smaller α?

2. For part 3, is the design and process method of the 2x2 splitter different from that of the 1x2 couplers? If so, what is the significance of this comparison? Please explain the purpose of this comparison/difference so to protect the coherence of the whole manuscript.

3. Equations 1-3 and Equation 4-6 may overlap in form. Can these evolve into a unified set of formulas?

4. There is a typo in Table 1: 14890-1590.

Reviewer 3 Report

Comments and Suggestions for Authors

Interesting and well organized work, including design, fabrication and test of new devices.

I only suggest authors to include a discussion about the (at least expected) temperature dependence of performances (e.g. in terms of losses, splitting ratio stability, bandwidth, ...) due to thermo-optic effect, compared to other more standard approaches to power splitting.

Some figures need fixing some bugs, such as Fig. 3d 3e (and several others), where I see nothing leading "=1550 nm";

or Fig. 4 and Fig 8, where I do not see the "micro" symbol following lengths (2, 3 , 10 um).

Reviewer 4 Report

Comments and Suggestions for Authors

The manuscript presents a continuation of the work referenced in [53], expanding upon their prior research. In this study, the authors introduce the concept of utilizing polynomial-based adiabatic taper profiles to optimize the design of 1x2 and 2x2 power splitters. The considered designs are not only compact in size but also compatible with CMOS foundry processes, particularly concerning minimum dimensions.

The manuscript follows a structured approach, commencing with simulation-based optimization and culminating in experimental validation. This systematic progression enhances the reader's comprehension and lends credibility to the findings.

Overall, I do not see any major mistakes and it could be almost accepted as it is. However, one point of confusion arises regarding the choice of a slab waveguide for the 2x2 power splitter. The rationale behind this selection is either inadequately explained or not readily discernible.

Minor confusions/comments:

1) Line 212 - It might be better to write a reasoning;

2) Chapter 2.3 second paragraph. It might be better to include also a Figure regarding the experimental setup;

3) Line 247 - before "," might be better to mention over all of the 9 measured designs;

4) Line 346 - 2 sentences in row start with "The selected p_n values are" should be improved.

5) Line 406 - Replace "  " to " " (2x spaces typo).
